


# Exploring Holocene temperature trends and a potential summer bias in simulations and reconstructions

Christian Wirths[1,2], Elisa Ziegler[3], and Kira Rehfeld[3]

[1]Climate and Environmental Physics, University of Bern, Bern, Switzerland
[2]Oeschger Centre for Climate Change Research, University of Bern, Bern, Switzerland
[3]Department of Geosciences and Department of Physics, University of Tübingen, Tübingen, Germany

**Correspondence:** Christian Wirths (christian.wirths@unibe.ch)

**Abstract.** Proxy-based reconstructions and climate model simulations of surface temperature trends during the Holocene disagree: While reconstructions show a cooling during the mid- and late Holocene, climate models show a continuous warming - a contradiction known as the Holocene temperature conundrum. Despite extensive research, the reason for the disagreement remains unclear. Both, missing processes in the models as well as biases in the proxies and the resulting reconstructions are possible sources of the conundrum. Here we compare our TransEBM v1.2 climate simulation as well as additional climate models of different complexity and Holocene temperature trends from the Temperature12k dataset (Kaufman et al., 2020b), with regards to model-data and model-model agreement. We show that models of all complexities disagree with mid-Holocene temperature trends in reconstructions and that this disagreement is almost independent of proxy and archive type. While, models show the highest agreement with summer temperature trends in reconstructions, our study shows that a trivial summer bias in proxies is not sufficient to explain the conundrum. Further effort to disentangle seasonal biases in proxies and the testing of potential misrepresentations in climate models, like anthropogenic land-use, in form of sensitivity experiments are needed to resolve the Holocene conundrum.

## 1 Introduction

Major questions about the climate evolution of the Holocene remain, despite its proximity in time. Climate reconstructions show an early Holocene warming period which peaks during the mid-Holocene (cmp. Fig. 1). The exact timing of it depends on the reconstruction, the proxy used and the applied methodology (Kaufman et al., 2020a; Cartapanis et al., 2022). After this Holocene climate optimum, global temperatures tend to decrease again until the start of the anthropogenic warming in the 19th century (Marcott et al., 2013; Marsicek et al., 2018; Kaufman et al., 2020a). On the other hand, climate models suggest a steady warming trend over the majority of the Holocene that follows the increasing greenhouse gas (GHG) concentrations and the retreat of the ice shields (Liu et al., 2014; Harrison et al., 2014; Brierley et al., 2020). This discrepancy is commonly known as the Holocene temperature conundrum (Liu et al., 2014).

Recent studies covering this topic proposed different approaches to reconcile the conundrum. Liu et al. (2014) and Erb et al. (2022, preprint) argue that part of the divergence between reconstructions and models is due to seasonal biases in the





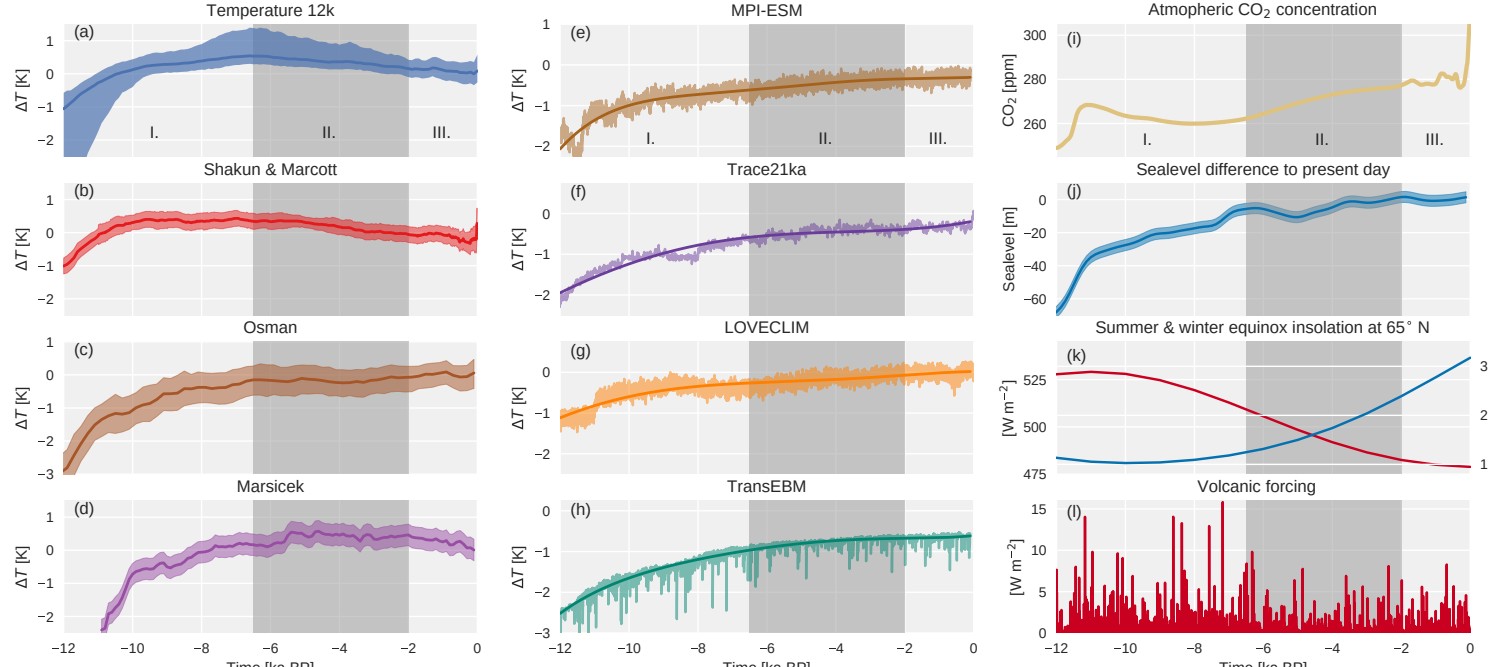

**Figure 1.** Annual Holocene temperatures in ka before 1950 (BP) from proxies (a-d), climate model simulations (e-h), and important boundary conditions (i-l). Proxy reconstructions from (a) Kaufman et al. (2020a), (b) Marcott et al. (2013); Shakun et al. (2012), (c) data assimilation from Osman et al. (2021), (d) pollen reconstructions for North America and Europe (Marsicek et al., 2018). Simulated Holocene temperatures from (e) MPI-ESM (Kleinen et al., 2020), (f) Trace21ka (Liu et al., 2009), (g) LOVECLIM (Timm and Timmermann, 2007), and TransEBM and the relevant climatic forcings (i) atmospheric $CO_2$ concentration (Köhler et al., 2017), (j) sealevel change (Grant et al., 2012), (k) summer (red) and winter (blue) equinox insolation at 65°N (Berger, 1978), and volcanic dimming (Sigl et al., 2021; Toohey et al., 2022, in prep.). I, II, III mark the choosen time periods for the early (12-6.5 ka BP), mid (6.5-2 ka BP) and late (2-0.1 ka BP) Holocene. Proxy records show a more or less pronounced cooling during the mid- and late Holocene, while models show a continuous warming following the increasing $CO_2$ concentrations and retreating ice sheets.

proxies. A seasonal bias might occur when a paleoclimate archive is more sensitive to changes during a specific season than to changes of the annual mean. Consequently, a seasonal change, for example in summer insolation, would then be imprinted into the archive and could be misinterpreted as an annual signal. In support of this, Bova et al. (2021) suggest that Holocene sea surface temperatures (SST) might have a substantial seasonal summer bias. SST reconstructions from alkenone and Mg/Ca in particular are affected by a seasonal bias that depends on latitude due to the seasonality of productivity rates (Christiansen and Ljungqvist, 2017; Leduc et al., 2010; Schneider et al., 2010). Likewise pollen-based temperature reconstructions are suspected





to contain an imprint from summer temperatures (Rehfeld et al., 2016). Those seasonal biases can be tackled by using expert knowledge of the individual sites to attribute the individual reconstructions to a season (Kaufman et al., 2020b). However, this method is limited by the current knowledge of the archives ecology (Kaufman et al., 2020b). Yet, overlooked summer biases in proxies might not be sufficient to explain the large difference between reconstructions and models.

35

Therefore Erb et al. (2022, preprint) suggest that the Holocene temperature conundrum might be the result of multiple interacting factors. Another line of evidence suggests that the cooling in North American and European pollen records only happens late in the Holocene (Marsicek et al., 2018). Similar results were found by Kaufman et al. (2020a), who showed that the timing and magnitude of the Holocene temperature maximum strongly depends on the chosen region and reconstruction method. In addition, Cartapanis et al. (2022) find a difference in magnitude and timing of the Holocene temperature maximum between marine and land-based proxies. Most recently, Osman et al. (2021) used a data assimilation approach, to show that the reconstructed mid- to late Holocene cooling might be a feature of poor spatial coverage and averaging methods.

With respect to climate model simulations, Bader et al. (2020) found that both warming and cooling climate modes coexist during the Holocene. In their simulations, this produces a short and relatively small late Holocene cooling. Others argue that models lack key processes, thus failing to simulate important changes of the climate system during the Holocene, for example due to a low orbital climate sensitivity and underrepresented feedback processes in many models (Liu et al., 2014). Although volcanic forcing alone would not be sufficient to induce a strong cooling in the models it might still provide part of the explanation (Ellerhoff et al., 2022). In addition, including Holocene anthropogenic land-use might be another step to simulate the Holocene climate more accurately (Liu et al., 2014; Harrison et al., 2020).

While a lot of explanations have been proposed, no clear answer has emerged. Therefore, we further investigate differences of Holocene climate model simulations and temperature reconstructions. We compare the capability of the 2D energy balance model TransEBM v.1.2 and climate models of higher complexity to reproduce reconstructed Holocene temperature trends targeting four major questions. First, does the ability of climate models to reproduce reconstructed temperatures change over time during the Holocene? Second, are there differences in annual and seasonal comparisons between model simulations and reconstructions? Third, can we identify regions and proxies of higher and lower agreement with model simulations? And finally, we discuss how the model complexity influences the outcome of the investigation.

## 2 Data and Methods

We analyze simulations from climate models of varying complexity: Starting with TransEBM (Ziegler and Rehfeld, 2020), a 2D energy balance model that simulates the linear response of the climate system via the intermediate complexity model LOVECLIM (SIM2bl Timm and Timmermann, 2007) to the Earth System Models (ESMs) CCSM3 (Trace21ka Liu et al., 2009) and MPI-ESM 1.2 (Kleinen et al., 2020). The main features and forcings of the simulations are summarized in Table



1. We extended the model version of TransEBM v.1.0 (Ziegler and Rehfeld, 2020) to include freshwater and zonal volcanic
65  forcing. The model is initialized at 26 ka BP and applies $CO_2$ (Köhler et al., 2017), orbital (Berger, 1978), volcanic (Sigl et al.,
2021; Toohey et al., 2022, in prep.), solar, sea-ice, land-ice (Peltier et al., 2015), and sea-level (Grant et al., 2012) forcing
to simulate the transient climate evolution until present-day. Sea-ice extent is linearly interpolated between the Last Glacial
Maximum (LGM) and present-day states given in Zhuang et al. (2017). A more detailed description of the model and forcings
can be found in Section A of the Appendix.

70

We contrast TransEBM v1.2 with simulations from LOVECLIM (SIM2bl Timm and Timmermann, 2007), CCSM3 (Trace21ka
Liu et al., 2009), and MPI-ESM 1.2 (Kleinen et al., 2020). All simulations used in this study simulate the transient climate evo-
lution from LGM to present-day, thus covering the Holocene period of our study from 12 ka BP to 1850 AD. All simulations
include GHG forcing by Köhler et al. (2017), orbital forcing following Berger (1978), and include a dynamic-thermodynamic
75  sea-ice model. The ice-sheets in all simulations are externally forced, using different reconstructions (c.f. Table 1). The models
of higher complexity, MPI-ESM 1.2 and CCSM3 further couple dynamical vegetation models. Freshwater forcing is handled
by a river run-off hydrology model in MPI-ESM 1.2 and by prescribed meltwater discharge in Trace21ka.

Information on the Holocene temperature trajectory can also be inferred from paleoclimate archives. Here, we use the
80  Temperature12k multi-proxy dataset (Kaufman et al., 2020b), which includes 1319 ecological, biophysical and biochemical
paleo-temperature proxy records from 679 sites. Around 70% of those sides are land- and approximately 30% are marine-based
(Kaufman et al., 2020b). Pollen and chironomids are the most common proxies for land and alkenones and isotopes for marine
sites. 74% of the sites contain proxies representing annual temperatures, while 64% and 39% of the sites contain proxies repre-
senting seasonal summer and winter temperatures respectively (Kaufman et al., 2020b). On average the length of the Holocene
85  records is 9813 years (Kaufman et al., 2020b). The temporal resolution of 15% of the records is 50 years or finer, 39% have
a resolution of 51 to 150 years, 25% have a resolution between 151 and 250 years, and 21% have a resolution coarser than
250 years (Kaufman et al., 2020b). Spatially, 51% of the proxy sites are located between 30° and 60°N with the highest spatial
density over North America and Europe (Kaufman et al., 2020b). In contrast, only 16% of the proxy sites are located in the
Southern Hemisphere.


For both reconstructions and simulations we calculate annual and summer (JJA) temperature trends by performing a linear
regression for three time periods: Period I) from the early Holocene (12 ka BP) to the mid-Holocene (6.5 ka BP), including the
Holocene thermal maximum (Kaufman et al., 2020a). Period II) from the mid-Holocene to the late Holocene (2 ka BP) and
period III) from the late Holocene to the pre-Industrial (PI; 0 ka BP/ 1850 AD). We then classify the calculated trends into three
classes, warming, cooling and no significant trend, by using a T-test with a 5% significance level and assuming independent
data points. Finally, we define a measure of agreement between simulations and reconstructions, hereafter referred to as trend
agreement. We compute this measure at the proxy locations for six 30°-wide latitudinal bands. The large differences in the
sources of variability between proxies and simulations make reconstructions without a significant trend difficult to compare to



simulations. Thus, we only compare reconstructions with a significant temperature trend with the simulations. For these, we

compile the number of models that fall in the same temperature trend class as the reconstructions for every proxy location. Thereby, we identify areas of overall model-data agreement and areas of mismatch.

|  | MPI-ESM 1.2 (Kleinen et al., 2020) | Trace21ka (Liu et al., 2009; He, 2011) | LOVECLIM (Timm and Timmermann, 2007) | TransEBM v1.2 (Ziegler and Rehfeld, 2020), this study |
|---|---|---|---|---|
| Resolution | $3.75° \times 3.75°$ | $3.75° \times 3.75°$ | $5.5° \times 5.5°$ | $2.8° \times 2.8°$ |
| Atmosphere | ECHAM6.3 | CAM3 | ECBILT | 2D-EBM |
|  | 31 layers | 26 layers | 3 layers | surface only |
| Ocean | MPIOM1.6 | NCAR-POP | CLIO | - |
|  | 40 layers | 25 layers | 20 layers |  |
| Sea-ice | dynamic-thermodynamic | dynamic-thermodynamic | dynamic-thermodynamic | forced |
| Vegetation | JSBACH | CLM-DGVM | - | - |
|  | Reick et al. (2013) | Levis et al. (2004) |  |  |
| Ice | GLAC-1D | ICE-5G | ICE-4G | ICE-6G |
|  | Tarasov et al. (2012) | Peltier (2004) | Peltier (1998) | Peltier et al. (2015) |
| GHGs | $CO_2$, $CH_4$, $N_2O$ | $CO_2$, $CH_4$, $N_2O$ | $CO_2$, $CH_4$, $N_2O$ | $CO_2$ |
|  | Köhler et al. (2017) | Joos and Spahni (2008) | Indermühle et al. (1999) | Köhler et al. (2017) |
|  |  |  | Smith et al. (1999) |  |
| Orbital | Berger (1978) | Berger (1978) | Berger (1978) | Berger (1978) |
| Freshwater | river run-off | meltwater discharge | river run-off | - |

**Table 1.** Summary of the model configurations of the MPI-ESM (Kleinen et al., 2020), Trace21ka (Liu et al., 2009), LOVECLIM (Timm and Timmermann, 2007), and TransEBM simulations and their applied climatic forcings.

## 3 Results

### 3.1 Model-data comparison of annual temperature trends

The annual temperature trends of the models and reconstructions are depicted in Fig. 2 and 3 for all three time periods. For the early Holocene, all models simulate a warming trend that is especially strong in the Northern Hemisphere and Antarctica. LOVECLIM and MPI-ESM show an additional cooling trend in the North Pacific. Generally, cooling appears over the oceans, with the exception of LOVECLIM which simulates cooling also in eastern Siberia. TransEBM is the only model which exclusively shows warming in all regions. The proxy reconstruction trends (c.f. Fig. 2) primarily indicate warming (51.1%) as in

the simulations. This is especially pronounced in the North Atlantic, North America and Europe. There are numerous spatially





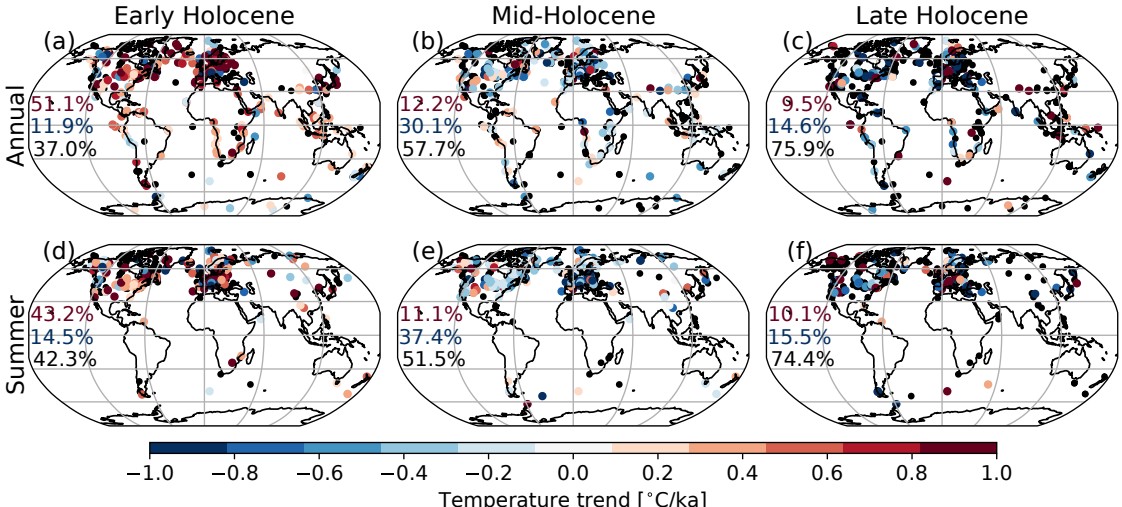

**Figure 2.** Reconstructed annual (a, b, c) and summer (d, e, f) temperature trends for the early Holocene (12-6.5 ka BP), mid-Holocene (6.5-2 ka BP), and late Holocene (2-0.2 ka BP). The numbers indicate the percentage of reconstructions with a warming (red), cooling (blue) or no significant trend (black). During the early Holocene the majority of reconstructions show a warming trend. For the mid- and late Holocene the majority of reconstructions show no significant trend. Most of the reconstructions with a significant trend during these periods indicate cooling.

distributed locations with cooling trends (11.9%), often in the direct neighborhood of reconstructions that show warming. Such cooling is found both in terrestrial and marine records.

During the mid- Holocene the models simulate warming in most regions (c.f. Fig. 3). Yet, all models also show areas of cooling. The MPI-ESM, LOVECLIM and Trace21ka simulations show some cooling in high northern latitudes as well as scat-
tered over land in the lower latitudes. TransEBM alone simulates cooling over Antarctica and only there. The reconstructions, on the other hand, show no significant trend in 57.7% of the locations and cooling in 30.1% of them. However, 12.2% of the reconstructions show a warming trend. Most of those reconstructions are located in North America, Europe and Southeast Asia.

During the late Holocene the models exhibit a warming band around the equator and subtropics and cooling outside, most
pronounced in TransEBM's linear response. MPI-ESM, too, indicates cooling in the higher latitudes. LOVECLIM shows cooling over the North Atlantic and Antarctica, while the cooling trends in Trace21ka are spread out over the South Atlantic and Southern Ocean as well as northern Eurasia and the Arctic Ocean. The reconstructions show no significant trend for 75.9 % of the locations. Warming and cooling trends exists for 9.5% and 14.6% of the proxy locations respectively, with most warming trends found in western North America, Southeast and East Asia.




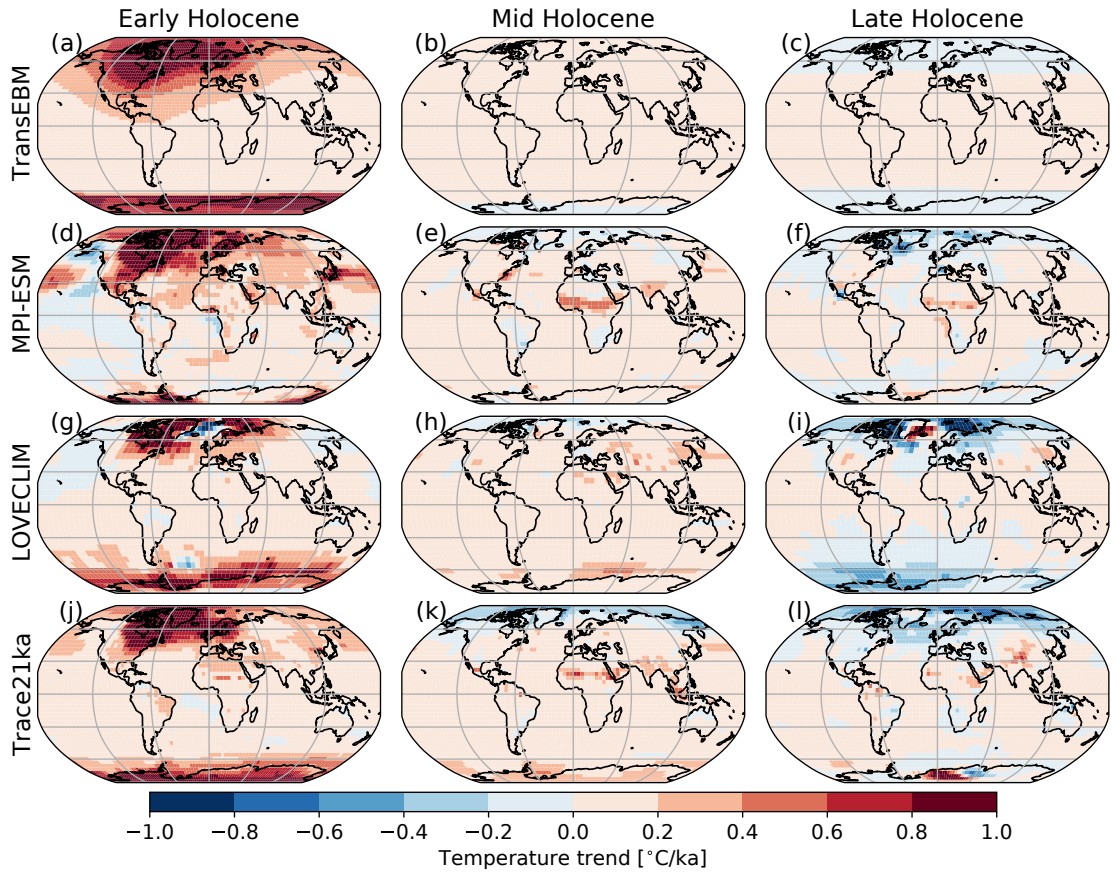

**Figure 3.** Annual model temperature trends for the early (12-6.5 ka BP), mid- (6.5-2 ka BP), and late (2-0.2 ka BP) Holocene. There is a general warming trend, especially in the high latitudes during the early Holocene. During the mid- and late Holocene, the warming trends become smaller and high altitude cooling occurs during the late Holocene.

## 3.2 Model-data comparison of summer temperature trends

The simulated summer temperatures depicted in Fig. 4 indicate warming in the southern mid- and high latitudes during the early Holocene. In the Northern Hemisphere there is a strong warming trend over the greater Hudson Bay area and Greenland, expect for TransEBM which shows a high latitude cooling and warming over the North Atlantic and the North American East Coast. All models simulate a pronounced continental cooling over Eurasia and, with the exception of TransEBM, light cooling over the oceans. In contrast, TransEBM shows warming trends over all oceans and the strongest warming in Antarctica.




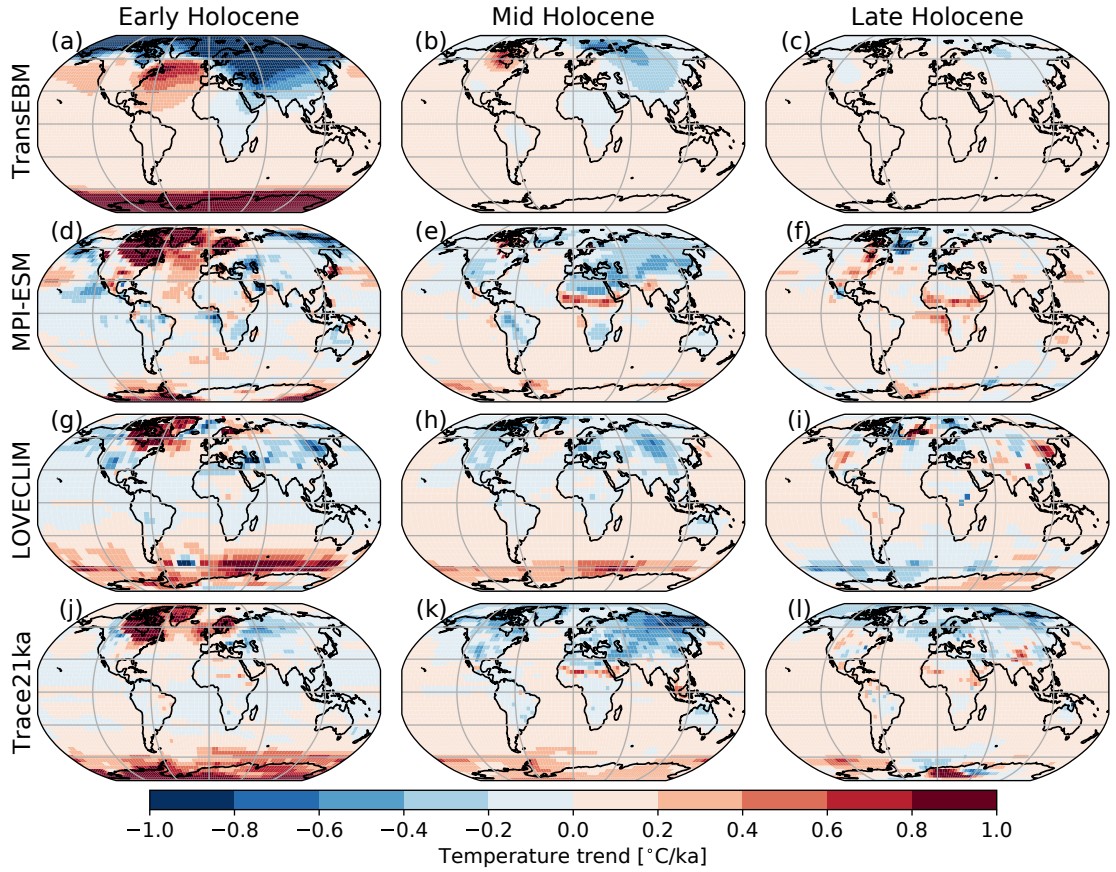

**Figure 4.** Summer model temperature trends for the early (12-6.5 ka BP), mid- (6.5-2 ka BP), and late (III: 2-0.2 ka BP) Holocene. There is a continental cooling trend during the Holocene, most pronounced in the Northern Hemisphere. During the early Holocene, a strong warming trend in the North Atlantic and eastern North America can be observed. Ocean temperatures show warming and cooling trends during the early and mostly warming trends during the mid- and late Holocene.

The reconstructions, on the other hand, show a warming pattern for 43.2%, cooling for 14.5% and no significant trend for 42.3% of the locations. The warming trend is most pronounced in North America and Europe. In the Southern Hemisphere the number of summer temperature reconstructions is low at only ten but the majority indicate warming. During the mid-Holocene, all models simulate cooling over the continents and the Arctic, while there is warming over Antarctica and the oceans. Reconstructions, on the other hand, mostly show a non-significant trend in 51.5%, cooling in 37.4%, and warming in 11.1% of the locations. Most reconstructions are located in the Northern Hemisphere with a cluster showing warming trends in the north west of North America. For the late Holocene cooling in the models is limited to the higher latitudes and



mostly continental areas. MPI-ESM and Trace21ka also show cooling over Antarctica. Everywhere else the simulations show warming. In the reconstructions, the number of non-significant trends increases to 74.4% of the locations, while the number of negative temperature trends decreases to 15.5% of the locations. The number of proxy locations with positive reconstructed temperature trends only decreases a little to 10.1%, most of them in north west North America and Central Europe.

While reconstructions tend to show similar patterns in annual and summer temperature trends over all periods, the models
simulate significant differences between summer and annual temperature trends, especially in the early and mid-Holocene.

## 3.3   Model skills at matching reconstructed trends for annual and summer temperatures

Comparing the annual latitudinal trend agreements between the individual simulations and reconstructions (c.f. Fig. 5 a-f) shows that the models' skills to fit the reconstruction trends is highest for the early Holocene and lowest during the mid-
Holocene. The trend agreement constantly decreases only for the southern high latitudes. Overall, a V-shaped pattern of a decrease followed by an increase in the trend agreement can be observed in all other latitudinal bands with the exception of LOVECLIM and MPI-ESM in the northern low latitudes. There, those two simulations show a continuous decrease in trend agreement. MPI-ESM further shows a flat trend agreement during the mid- and late Holocene in the northern high and mid-latitudes. Major differences between the model annual trend agreements mostly occur during the late Holocene. MPI-ESM
shows the lowest trend agreement during the late Holocene for all latitudinal bands in addition to the above mentioned features. Additionally, LOVECLIM and Trace21ka tend to compare better to reconstructions during the late Holocene than TransEBM and MPI-ESM.

For the summer temperature trends, depicted in Fig. 5, most proxies are located in the northern high and mid-latitudes.
There, the trend agreement increases from the early to the mid- Holocene for Trace21ka, LOVECLIM, and MPI-ESM. For TransEBM, the trend agreement decreases between those periods. For the late Holocene the trend agreement increases again for TransEBM while it decreases in all other models. In the southern mid-latitudes a similar picture emerges, although for MPI-ESM and Trace21ka the increase in trend agreement is only small or insignificant. For the northern low and the southern low and high latitudes, the abundance of reconstructions with significant trends is smaller than 3 during the entire Holocene.
Thus, it is impossible to make statements on these regions.

The comparison of annual proxy-based reconstructions and summer model trends shown in Fig. 5, tests whether annual reconstructions have a summer bias. The highest agreement between reconstructions and simulations occurs during the mid-Holocene, except for TransEBM. In particular, this agreement is higher than the one between annual proxies and simulations. However, during the early and late Holocene, the agreement between annual reconstructions and summer simulations is smaller
than the agreement between annual reconstructions and the annual model output.

Figure 6 shows how many models agree with the reconstructions at each proxy location. It indicates that in the early Holocene most models agree with reconstructions for the annual trend. For the mid-Holocene the agreement decreases, especially in North America, but also in Europe and East Africa. For the late Holocene the agreement between models and reconstructions



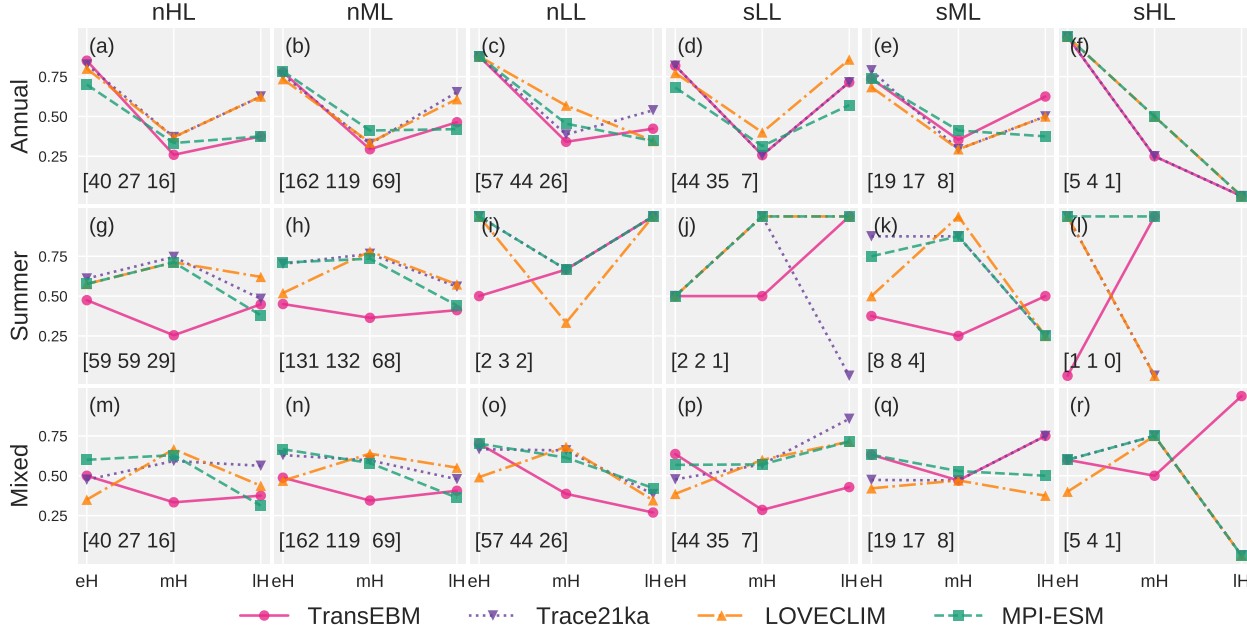

**Figure 5.** Degree of agreement between annual reconstructed and annual model (Annual), summer reconstructed and summer model (Summer), and annual reconstructed and summer model (Mixed) temperature trends over the early (eH 12-6.5 ka BP), mid- (mH 6.5-2 ka BP), and late (lH 2-0.2 ka BP) Holocene. The number of considered reconstructions for every time period is indicated in brackets. The trend agreement is separated into latitudinal bands as follows: northern high latitudes (nHL), northern mid-latitudes (nML), northern low latitudes (nLL), southern low latitudes (sLL), southern mid-latitudes (sML), and southern high latitudes (sHL). Annual reconstructions and model simulations show the lowest trend agreement during the mid-Holocene, whereas most summer reconstructions and simulations show highest trend agreement during that period. The mixed reconstructions and model simulations show higher trend agreement during the mid-Holocene than the annual reconstructions and simulations but lower trend agreement during the rest of the Holocene.

increases again, but models tend to agree less with each other than during the early Holocene. For the summer temperature
trends, there is a similar behavior as in the annual trends during the early and late Holocene. In contrast, the summer trends
during the mid-Holocene show more models agreeing with reconstructions in the Northern Hemisphere than for the annual
trends.

## 4   Discussion

The strong overall annual warming trend in models during the early Holocene (cf. Fig. 3) coincides with the increase of GHG
concentration and the retreat of the Laurentide ice sheet as indicated in Fig. 1. While models for the early Holocene agree with
over 75% of the significant proxies, for the mid-Holocene, the agreement decreases too below 35% as expected by the Holocene





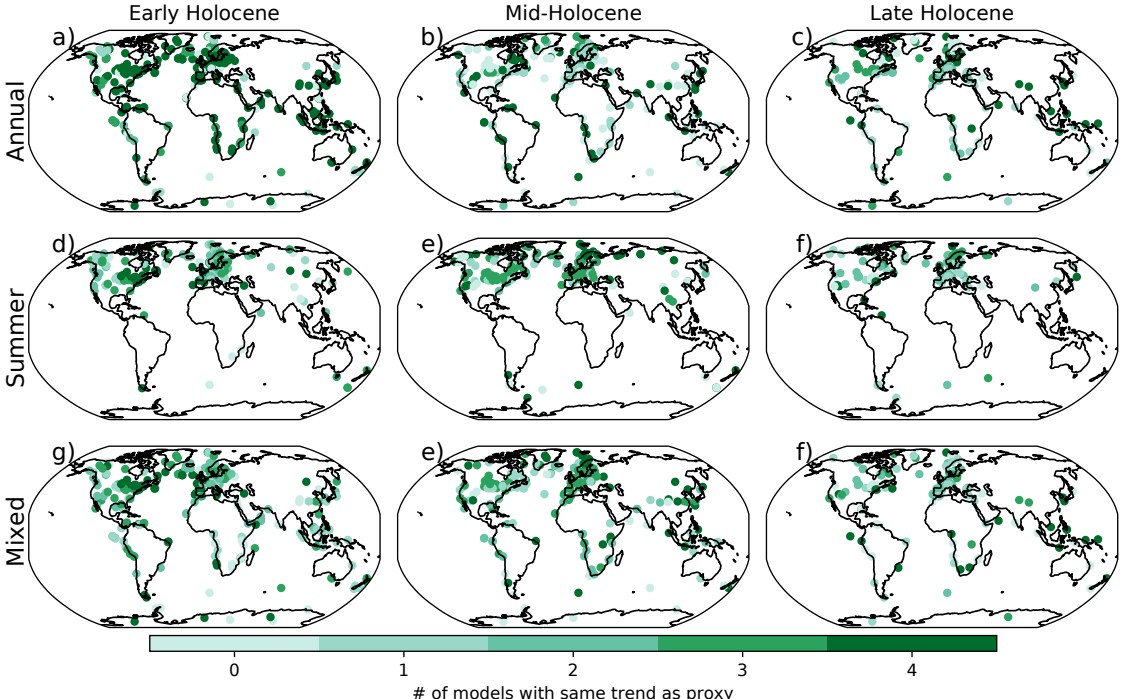

**Figure 6.** Number of models with the same temperature trend direction (cooling/warming) as the individual proxy record for the three time periods. Top row shows the match with simulated annual temperatures. Center row shows the match with summer temperatures. The bottom row shows the match when comparing annual reconstructions and summer simulations.

temperature conundrum (Liu et al., 2014). Note that we do not differentiate between a small and a large temperature trend as long as they are significant. This means that a high trend agreement does not always mean that the models really reproduce the reconstructed temperature well, but rather that they show similar trends in the temperature. In contrast to previous suggestions by Marsicek et al. (2018), our results do not support the claim that the conundrum only affects one specific region, since a mid-Holocene trend agreement minimum can be observed in every latitudinal band except the southern high latitudes.

### 4.1 Potential proxy biases

In contrast to the annual temperature trends, the summer trends show a mid-Holocene trend agreement maximum in regions of high proxy availability, especially North America and Europe. Those regions have large amount of pollen proxies (cmp. Kaufman et al. (2020a) Fig. 2 a)), a proxy known for potential biases towards summer months (Rehfeld et al., 2016). Additionally, annual and summer temperature trends in reconstructions are similar, while there are large differences between those trends among models. All of this suggests that the proxy-based reconstructions might be biased towards the summer months





as suggested before Liu et al. (2014); Rehfeld et al. (2016); Bova et al. (2021); Erb et al. (2022). To test this hypothesis, we
compare annual reconstruction and summer model temperature trends. The trend agreement between annual reconstructions
and summer model temperature trends shown in Fig. 5 does indicate a higher trend agreement than for annual reconstructions
and models during the mid-Holocene. However, for the early and late Holocene this is not generally the case. Hence, a potential
proxy summer bias seems to be of a more complex nature. An explanation why only the mid-Holocene might be affected by a
strong seasonal bias in the reconstructed temperature trends lies with the forcings (c.f. Fig. 1) during the Holocene: While the
northern summer insolation already decreases during the early Holocene, the increase in atmospheric carbon dioxide and the
retreat of the ice sheets are the dominant drivers of the warming climate. During the mid-Holocene, however, changes in the
ice sheets are small while the decrease in northern summer insolation is still strong. Thus, a summer bias emerges especially
during that period. During the late Holocene changes in the summer insolation are small such that a strong summer bias can
not be expected.

The separate analysis of reconstructions from marine and non-marine proxies show similar results (c.f. Fig. B1 andB2),
but if a summer bias affects marine records for temperature as described by Bova et al. (2021), it appears to be smaller than
for non-marine proxies in our results. Our study indicates that the northern high and mid-latitudes are most influenced by a
seasonal bias, due to the large fraction of non-marine, mostly pollen proxies. In contrast, marine proxies do not show pro-
nounced regional differences due to the weaker signal of the seasonal bias. While a summer bias in the proxies might explain
the mid-Holocene model-data differences in the northern high and mid latitudes, it leaves many questions regarding the other
regions. A more complex bias towards different seasons might be an explanation, an idea which is supported by findings of
Bova et al. (2021), who showed that tropical SST reconstructions show biases towards different months depending on proxy
and location. To make progress in this direction it is important to deepen our understanding about various drivers and their
impact on paleoclimatic archives and the related proxy system.

## 4.2 Are the models at fault?

The widespread disagreement during the mid-Holocene over the entire range of model complexity constrains potential faults on
the modeling side. Assuming that the proxy-based reconstructions are right, all models either miss a certain process completely
or a process is parameterized wrong in all models. A possibility would be an orbital climate sensitivity that is too low (Liu
et al., 2014). Further, it would be possible that the simulations are initialized with the wrong boundary conditions. For example,
a slow responding system like the deep ocean might be initialized wrong and limit Holocene warming in simulations. Evidence
that the ocean modelling requires improvement comes from reconstructions of mean ocean temperature (Bereiter et al., 2018;
Haeberli et al., 2021), which show a temperature peak at the onset of the Holocene. Additionally, the timing of the Holocene
temperature maximum in reconstructions from marine proxies at 10 to 7 ka BP shows large differences between the Northern
and Southern Hemisphere, providing further evidence for issues in the simulated Holocene ocean (Cartapanis et al., 2022).

The different timing of the Holocene temperature maximum between reconstructions from land- and marine-based proxies
might also indicate that the response of the vegetation system to the transient disequilibrium of the vegetation-climate sys-



tem during the Deglaciation is not properly represented in the models so far (Cartapanis et al., 2022). Unlike the decrease in agreement from early to mid-Holocene for annual temperatures, the comparison of the summer temperatures indicates that the

models have the same or better skill in simulating mid-Holocene summer temperature trends. Here, TransEBM shows the lowest trend agreement since its seasonal temperature pattern is, due to the linear nature of the EBM, almost exclusively driven by northern summer insolation (Wirths et al., 2021). This indicates that feedback mechanisms are of high importance to correctly simulate Holocene summer temperatures. The better mid-Holocene performance of the models for summer temperatures is especially apparent in Fig. 6, notably in North America and Europe.


Another implication from this study is the consistency with which models are able or unable to reproduce reconstructed temperature trend patterns. For the early and mid-Holocene, the simulations all show similar levels of agreement with the reconstructions: The model trends are either consistent with reconstructions for all models or do not fit the reconstructions at all. For the late Holocene, however, the models show a split picture, some models fit reconstructed trends, others do not.

A similar but less pronounced image persist for the summer months, with the exception that TransEBM is often not in line with the other models. It seems that the linear climate response given by TransEBM is not as sufficient for seasonal as for annual temperatures. In this regard, it might be of interest to investigate which individual model processes and feedbacks are important for seasonal but not as important for annual temperature evolution. One potential candidate might be the lack of a dynamical vegetation and land surface model in TransEBM, since TransEBM especially lacks the summer cooling patterns

over Europe and North America observed in the other models during the mid-Holocene. However, this might not be the only reasons since LOVECLIM also does not include a dynamical vegetation model but shows those cooling trends. Hence, missing representation of soil moisture and the latent heat flux as well as insufficient energy transport in TransEBM might compound the problem. Overall, no model shows an outstanding trend agreement over all periods and latitudes, showing that the complexity of the model and the ability to correctly simulate Holocene temperature trends are not trivially connected (Brierley et al.,

2020). Yet, for the northern high and mid-latitudes Trace21ka and LOVECLIM tend to show the highest agreement. A tool to gain a higher understanding of differences between proxy-based reconstructions and simulations are proxy system models. However, we require further advanced process-based understanding of the climate-archive-proxy-measurement chain to derive mechanistic proxy system models, adequate metadata for considered sites, and model simulation output for relevant variables (e.g. monthly thermocline depth, or growing-degree days) to rigorously test this hypothesis.


## 4.3 Lack of proxy coverage in the Southern Hemisphere

A major shortcoming of this kind of analysis is the low proxy coverage in the Southern Hemisphere leading to a high dependence on a small number of proxy records which are not necessarily representative for the region. Additionally, the proxy distribution within latitudinal bands is also heterogeneous. For example, the northern mid- and high latitudes mostly contain

records from North America and Europe while proxies are rather sparse in Eastern Eurasia. Due to the large differences in the source of variability between reconstructions and models, reconstructions without a significant trend were not used further





for model-data comparison, which reduces the abundance of records. This results in low proxy abundance in the Southern Hemisphere and during summer months. Further, reconstructions from shorter time periods like the late Holocene might have fewer data points which makes it more difficult to statistically determine significant trends.

## 5 Conclusion

This study investigated temperature trends throughout the Holocene in paleo-climate archives and climate model simulations. Models, with the exception of TransEBM, simulated similar annual and seasonal temperature trends. Annual temperatures showed mainly warming and only little cooling in the northern high latitudes, while summer temperatures showed a more pronounced cooling over the continents. Annual reconstructions and model simulations agree most during the early and late

Holocene and strongly disagree during the mid-Holocene. In contrast, reconstructions attributed to represent summer temperature, agree the most with the majority of models during the mid-Holocene and show lower agreement for the early and late Holocene. This effect holds for marine- and land-based proxies individually.

Regarding the Holocene conundrum, it follows that a simple seasonal proxy bias is unlikely as a full explanation. A bias in

proxy data might therefore depend on proxy type and location and the observed period. Further, our results indicate that models might incorrectly simulate parts of the Holocene climate. We suggest to investigate seasonal biases for individual proxy types, as done by Bova et al. (2021) for tropical SST reconstructions. To further rigorously test our hypotheses by utilizing proxy system models, a deeper process-based understanding of the proxy system as well as a higher abundance of metadata for testing is necessary. In addition, we found no trivial relationship between a model's complexity and the ability to reproduce

reconstructed Holocene temperature trends. Missing or biased boundary conditions and parametrizations in land-cover, ocean state and orbital climate sensitivity could lead to errors in the simulations. Additionally, anthropogenic land use might have a stronger influence than previously assumed (Harrison et al., 2020; Hopcroft and Valdes, 2021, 2022). Disentangling seasonal biases in proxy-based reconstructions as well as testing model improvements in sensitivity experiments might therefore bring us closer to a solution of the Holocene temperature conundrum.

*Code and data availability.* The Temperature12k was downloaded through WDS-NOAA Paleoclimatology (https://doi.org/10.25921/4RY2-G808). TransEBM v1.2 model code is available on Github (https://github.com/christianwirths/TransEBM1.2) as well as on Zenodo (10.5281/zenodo.7346941). The simulation results of TransEBM v1.2 simulation as well as the code to reproduce the analysis and plots of this publication can be found on Zenodo (10.5281/zenodo.7346941)(Wirths et al.).

## Appendix A: The TransEBM simulation

The model simulates the last 26 ka with several transient climate forcings: Carbon dioxide reconstructions from ice core records (Köhler et al., 2017) were used as annual greenhouse gas forcing. The orbital forcing is internally calculated by the model



following Berger (1978). Additionally, volcanic forcing from the Easy Volcanic Aerosol (EVA) forcing generator reconstructed by Toohey et al. (2022, in prep.) was annually averaged and regridded using linear interpolation. It was then applied for the latitudinal bands with a width of 2.8°of TransEBM. Annual solar forcing was artificially created from a reconstruction of
Holocene solar irradiance by Steinhilber et al. (2009) with a surrogate method: The reconstruction is based on [10]Be radionuclide measurements from ice cores and covers the last 9300 years. To increase the length of the time series of solar forcing, several artificial time series were constructed using a amplitude adjusted Fourier transform (AAFT) surrogate algorithm with random seeds. The land-sea mask and ice covers were created using reconstructions of the ice sheet and land-sea distribution by Peltier et al. (2015) spaced in 500-year intervals. Depending on the pace of the Deglaciation the ice sheets were interpolated between
two consecutive ice sheet reconstructions in accordance with the sea level change. For land-based ice the interpolation was performed by identifying the changes in the land-ice cover between two consecutive timesteps. Where ice cover changed, ice was gradually removed according to the sea level change, with grid cells closer to the equator removed first. If less ice needed to be removed than available for the same latitude, ice with the smaller longitude was removed first. This resulted in updates of the land-ice cover in 125 to 500 year intervals, depending on the pace of Deglaciation. The sea-ice was interpolated using
the same method. For sea-ice, the distributions given by Zhuang et al. (2017) were used as fix points for present-day and LGM conditions.

## Appendix B: Model-data comparison for marine and non-marine proxies and the effect of different reconstruction methods

The agreement between proxy and simulation temperature trends for marine and non marine-proxies are separately depicted
in Fig. B1 and Fig. B2. The comparison of annual temperature trends shows a strong decrease of agreement during the mid-Holocene for both marine and non-marine proxies as long as a sufficient number of proxies are available. For summer temperature trends, the number of marine proxies, which are supposed to represent summer temperatures, is larger than 5 only for the northern high latitudes. There, all models except TransEBM show higher mid-Holocene agreement than annual proxies and simulated temperatures. For non-marine proxies the same is found for the northern mid latitudes. For the mixed comparison
between annual proxies and simulated summer temperatures, the reconstructions from marine proxies show an inconclusive picture although the trend agreement is slightly higher during the mid-Holocene than in the annual comparison. For non-marine proxies, a substantially higher mid-Holocene trend agreement can be observed compared to the annual agreement in the northern high and mid-latitudes.





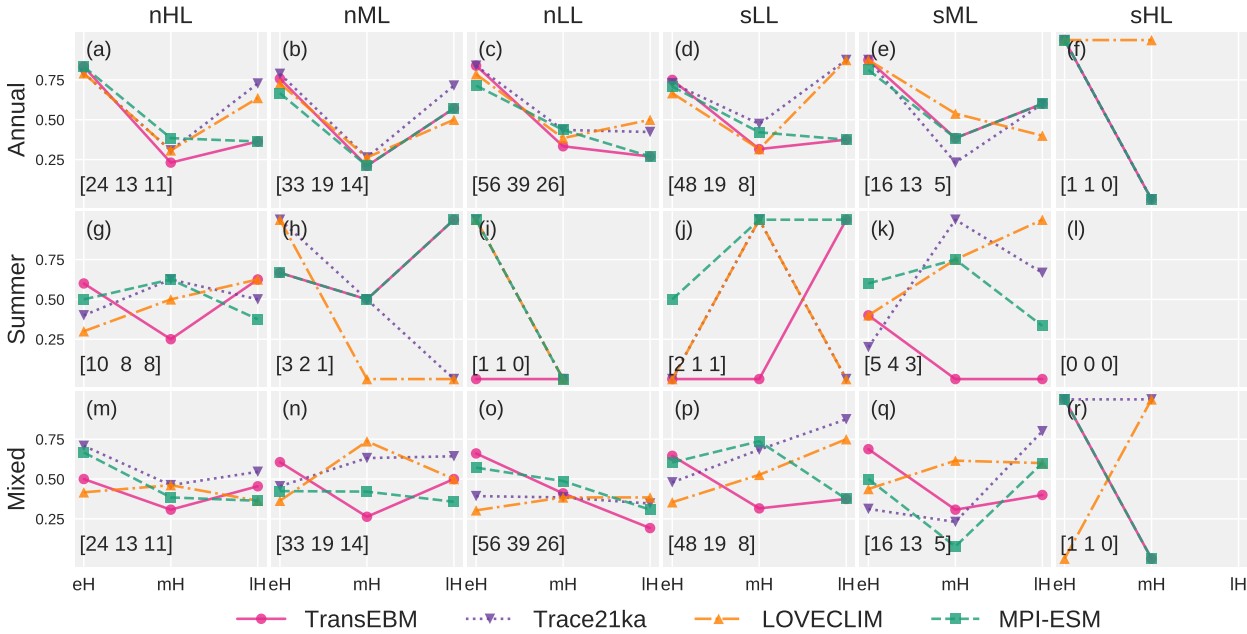

**Figure B1.** Degree of agreement between annual marine proxy and annual model (Annual), summer marine proxy and summer model (Summer), and annual marine proxy and summer model (Mixed) temperature trends over the early (eH 12-6.5 ka BP), mid- (mH 6.5-2 ka BP), and late (lH 2-0.2 ka BP) Holocene. The number of considered reconstructions for every time period is indicated in brackets. The trend agreement is shown separated into latitudinal bands as follows: northern high latitudes (nHL), northern mid latitudes (nML), northern low latitudes (nLL), southern low latitudes (sLL), southern mid-latitudes (sML), and southern high latitudes (sHL).



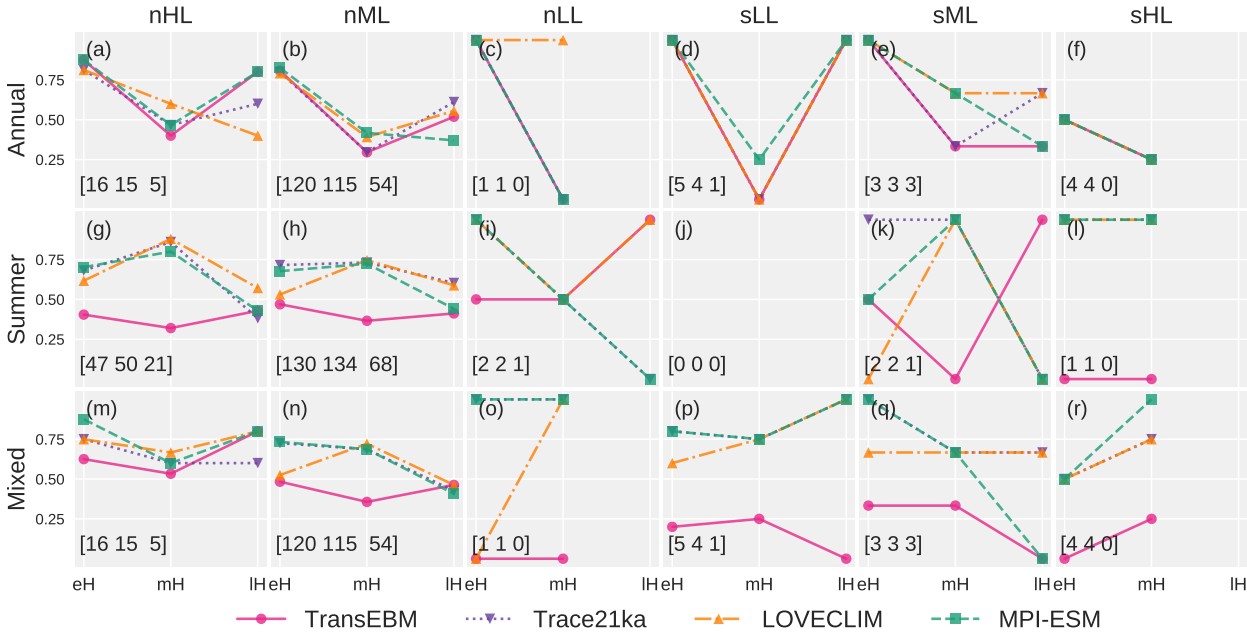

**Figure B2.** Degree of agreement between annual non-marine proxy and annual model (Annual), summer non-marine proxy and summer model (Summer), and annual non-marine proxy and summer model (Mixed) temperature trends over the early (eH 12-6.5 ka BP), mid- (mH 6.5-2 ka BP), and late (lH 2-0.2 ka BP) Holocene. The number of considered reconstructions for every time period is indicated in brackets. The trend agreement is shown separated into latitudinal bands as follows: northern high latitudes (nHL), northern mid latitudes (nML), northern low latitudes (nLL), southern low latitudes (sLL), southern mid-latitudes (sML), and southern high latitudes (sHL).




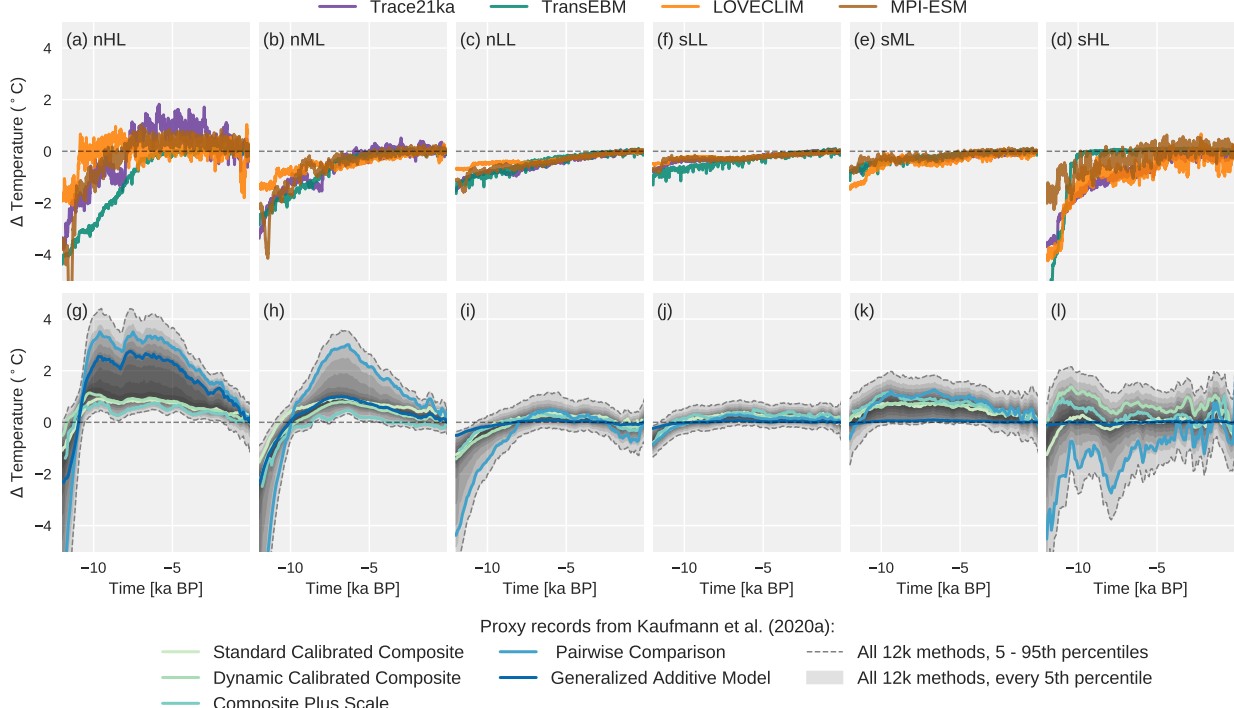

**Figure B3.** Temperature anomalies with respect to the period 1800-1900 C.E. For Trace21ka, TransEBM, LOVECLIM, and MPI-ESM (a-f) and proxy reconstructions (g-l) from Kaufman et al. (2020a) for six individual latitudinal bands. The model output was smoothed by fitting a fifth order polynomial. The proxy reconstructions show the median for five different methods (Standard Calibrated Composite (SCC), Dynamic Calibrated Composite (DCC), Composite Plus Scale (CPS), Pairwise Comparison (PAI) and Generalized Additive Model (GAM)), as well as the 5 - 95th percentile range of the ensemble over all methods. The strongest mid-Holocene cooling in the reconstructions occurs in the northern high and mid-latitudes for the PAI and CPS methods. The spread between different methods is also largest in the northern high and mid-latitudes. In contrast, only some models show a minimal cooling in the northern high latitudes, and no cooling or warming in the other latitudinal bands. Adapted from Kaufman et al. (2020a)



*Author contributions.* CW, EZ, and KR conceptualized this study, decided on the methodology, created the analyses, edited and reviewed

this manuscript. CW wrote the manuscript, implemented the software and computations and performed the visualization of the data, which

EZ and KR supervised.

*Competing interests.* The authors declare that there are no competing interests.

*Acknowledgements.* We acknowledge funding by the Swiss National Science Foundation through the pleistoCEP2 project (grant no. 200492

), the German Research foundation (project no. 395588486) as well as the German Federal Ministry of Education and Research (BMBF,

grant no. 01LP1926C, PalMod II). We further thank Thomas Stocker for comments on this manuscript and Jürgen Bader for discussion.



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
