# Peer review of "Exploring Holocene temperature trends and a potential summer bias in simulations and reconstructions"

_EGUsphere, 2023_

## Author Comment (AC1)

**Reply to comments by reviewer 1: Exploring Holocene temperature trends and a potential summer bias in simulations and reconstructions (egusphere-2023-86)**

Christian Wirths, Elisa Ziegler, Kira Rehfeld

**Summary of Changes**

We are grateful to the reviewer for evaluating our work, and the valuable and constructive comments that help substantially improve the manuscript. In response, we now

- implement text revisions throughout the manuscript, further detailed below in the response where appropriate
- expand the analysis by adding another simulation from HadCM3 of the Holocene
- expand on the role of the EBM in the analysis by providing and analysing sensitivity experiments; for analysing the role of volcanism we contrast them with an additional available simulation with HadCM3

We want to thank the reviewer again for the comments and suggested improvements of the manuscript. Below, we respond to the reviewer's individual comments in detail and describe the actions we will take to address them. We are convinced that our proposed actions will further improve the scientific value of our study.

**Detailed response**

(Original report cited in italics)

**Reviewer's comment:**

*The manuscript compares the temperature trends simulated by climate models of various complexities with paleo data over the Holocene. Many explanations have been suggested to explain the disagreements between models and data, the authors addressing specifically the impact of potential seasonal biases. They describe in details the seasonal and spatial distribution of the trends in the selected models and in data. This description is very clear. The paper is well written and easy to follow. I thus have no minor comment or suggestion to improve the presentation of the manuscript. However, there are two major points to consider in a revised version of the text.*

**Authors' reply:**

We thank the reviewer again for his constructive and helpful comments. We address the raised points below and propose actions to clarify the points and to further improve the manuscript.

**Reviewer's comment:**

*1/ The added value of the study is not clearly explained and the authors should insist more on this in the conclusion, which is very short in the current version of the manuscript. The first paragraph of the conclusion summarizes the description of the trends presented in the previous sections. The second (and final) paragraph starts by a quite mild sentence: 'Regarding the Holocene conundrum, it follows that a simple seasonal proxy bias is unlikely as a full explanation' and then present some general suggestions for improvements or new studies. The fact that seasonal proxy biases might play a role but could not explain the full model-data disagreement is already around for some time (see the recent review of Kaufmann and Broadman, 2023) and the authors should explain more clearly the new contribution they bring to the debate.*

**Authors' reply:**

We agree with the reviewer that in the current version of the manuscript and in the light of the recent review [Kaufmann and Broadman, 2023], which was not published at the time of submission, our conclusions can be sharpened to highlight the novelty of our study and how it further adds to the general discussion on Holocene temperature trends. We therefore plan to elaborate on our findings and contextualize them better in the revised version of the manuscript. Additionally, we will substantiate our conclusions further by broadening our simulation dataset and performing additional hypothesis testing as described below.

Action:

- We will implement text revisions throughout the manuscript to elaborate on the novelty of our study. In particular, this will be done in the discussion and conclusion.
- We will further support the analysis by providing and analysing sensitivity experiments, focussing on the role of volcanism in Holocene temperature development.

**Reviewer's comment:**

*2/ The authors analyze relatively old simulations that have been discussed in several studies. The selected data base has also already been used in model-data comparisons. A new simulation is included (TransEBM) but it has in general a lower agreement than the other ones with observations (see for instance Figure 5). This new simulation might be helpful to understand some of the characteristics of the other models but this is not developed in the current version of the manuscript. Furthermore, the set of selected experiments is not designed to test hypotheses, such as the potential role of vegetation or of the volcanic forcing for instance, as done in some other studies. Several transient Holocene have been performed recently. Some only cover parts of the Holocene or might not be publicly available but a larger set of experiments would provide additional information for the discussion (see for instance Askjær et al. 2022, in particular Fig. 3).*

**Authors' reply:**

We agree with the reviewer that in addition to the transient Holocene simulations used in our study there are more simulations on their way to publication or documented in publications. However, many of them cover only parts of the Holocene, which reduces their value for our study as we explicitly aim to compare models and reconstructions for the early, mid- and late Holocene. Simulations should, therefore, cover these time periods, which only few published simulations do and we consider almost all of those already. Aksjær et al. 2022 also includes simulations NNU12k and with HadCM3, which could extend the set of simulation we analyse. We therefore got in contact with the authors and have been granted access to the data from the HadCM3 simulations. Unfortunately, after several attempts with the involved authors, we have to conclude that the NNU12k simulations seems to be currently not available.

Furthermore, we agree with the reviewer that discussing the added value of TransEBM as a low complexity model should be expanded and improved in the manuscript. A major strength of low complexity models is their low computational cost such that they can be used for single forcing experiments and hypothesis testing, as mentioned by the reviewer. Especially the role of volcanism during the Holocene and its impact on Holocene climate is relatively unknown [Bader et. al. 2020]. Therefore, we will add analyses of additional sensitivity experiments we performed using TransEBM (cmp. Fig. 1) and those available for HadCM3 simulations to test the impact of volcanic forcing. By comparing TransEBM with HadCM3 simulations, we investigate the impacts of non-linear processes and feedbacks of the volcanic forcings on the Holocene climate.

[Figure]

Fig. 1: Temperature anomalies with respect to 12k BP for different sensitivity simulations of TransEBM. *Volcanic, ice sheet, sea ice, landmask, solar, orbital* and *CO2* are simulations in which only one forcing

component changes transiently while the other forcings are kept constant. All forcings change transiently in the *all* simulation.

Action:

- We will improve the discussion of TransEBM and its added value for our study throughout the text.
- We will include sensitivity experiments using TransEBM and HadCM3 to discuss the impact of volcanic forcing, in particular.

References:

Askjær et al., 2022. Multi-centennial Holocene climate variability in proxy records and transient model simulations. Quat. Sci. Rev. https://doi.org/10.1016/j.quascirev.2022.107801

Bader, J. et al. Global temperature modes shed light on the Holocene temperature conundrum. Nat. Commun. 11, 4726 (2020).

Kaufman D.S. and E. Broadman, 2023. Revisiting the Holocene global temperature conundrum. Nature 614, 425-435 . https://doi.org/10.1038/s41586-022-05536-w

---

## Author Comment (AC2)

**Reply to comments by reviewer 2: Exploring Holocene temperature trends and a potential summer bias in simulations and reconstructions (egusphere-2023-86)**

Christian Wirths, Elisa Ziegler, Kira Rehfeld

**Summary of Changes**

We thank the reviewer for taking the time to assess our work and providing valuable and detailed feedback. In response, we revise and expand the analysis by

- further exploring the role of seasonality in the temp12k dataset by,
- expanding on the role and benefits of TransEBM. To this end, we carry out revisions in the text and appendix and add sensitivity experiments,
- discussing the presence/ absence of the late Holocene cooling mode in different MPI-ESM simulations.

Below, we provide a detailed response to the reviewer's comments and describe the actions we plan to take in response. We are grateful for the reviewer's comments and suggested improvements, which will help improve our study as outlined below.

**Detailed response**

(Original report cited in italics)

**Reviewer's comment:**

*Wirths et al. summarise a comparison of simulations and reconstructions of temperature trends over three phases of the Holocene. They confirm important differences between the warming seen in models and the cooling reconstructed in temp12k. They conclude that a simple seasonal bias in the temp12k proxy records is unlikely to be able to explain the differences with the model runs.*

**Authors' reply:**

We thank the reviewer for the positive evaluation of our work.

**Reviewer's comment:**

*A main assumption within this work is that the seasonal and annual reconstructions extracted from temp12k are able to separately resolve the seasons. I am no expert in this dataset, but this seems unlikely to be the case. In fact if we can take the separate seasonal records at face value then we can immediately conclude that the seasonal bias is not the answer to the 'Holocene*

*temperature conundrum'. My best guess would be that the annual and summer reconstructions are probably both somewhat a mixture of several seasons, consistent with line 144: "reconstructions tend to show similar patterns in annual and summer temperature trends over all periods".*

Authors' reply:

We agree with the reviewer that the uncertainties attached to the seasonal reconstructions of the Temp12k dataset imply that they might not represent the season they are labelled as. Since the different proxies and reconstructions have a broad range of specific seasonality, the authors of the dataset (Kaufmann et. al. 2020b Scientific Data) used a "Season General" category to classify the reconstructions. They classify them in three general categories, annual, summer and winter, which we use in this study. Records representing spring or autumn were grouped with summer or winter, respectively (Kaufmann et. al. 2020b, Scientific Data). Additionally, Kaufmann et al. categorized proxy reconstructions supposed to represent a signal longer than six months, including June, as "annual" (Kaufmann et. al. 2020b, Scientific Data). A more detailed explanation of the seasonal labelling in the Temp12k dataset can be found in Kaufmann et. al. (2020b). We agree with the reviewer, that those decisions made by the original authors need to be addressed in more detail in our manuscript. We will therefore discuss this and the resulting implications in more detail in the revised version of the manuscript.

In our manuscript, we assume that the captured signal is dominated by the climatic conditions of the labelled season as it represents the best approximation of the recorded season according to the authors of the Temp12k database. However, we agree with the reviewer that proxy reconstructions labelled as summer do not necessarily represent the climate signal of the summer months. Instead they might be a convolution of the climate signal with a "productivity function". The use of proxy system models would be best able to account for this. As this is out of the scope of this study, we will add several tests to address these issues: First, in addition to checking for a summer bias in proxies, we will check whether certain proxy seasons might fit other model data better, e.g. compare trends in summer proxies with simulated winter temperatures. Second, we will compare model timeseries at proxy locations with each other, e.g. compare simulated winter and summer temperature trends. Since different proxies will exhibit different seasonal biases, this will establish upper bounds for possible effects of mislabelled seasons. Finally, we will test the effect of labelling proxy timeseries covering parts of the year (signals longer than six months, including June) as "annual", by analysing the effect of doing so in the simulations, e.g. comparing data from April to September with the annual timeseries.

Those additional analyses will allow us to provide upper limits on the influence of mislabelled seasons and allow us to quantify potential uncertainties in the classification.

Action:

- We will enhance the discussion of the Temp12k dataset, its labelling of different seasons and the underlying assumptions throughout the manuscript.

- We will carry out additional analyses to test and provide upper limits for the uncertainties attached to the seasonal labels.
- These additional results will be added to the manuscript and discussed.

**Reviewer's comment:**

*In the abstract it is stated that: "our study shows that a trivial summer bias in proxies is not sufficient to explain the conundrum". I believe this is based on the comparison of summer simulations with annual records but it's not clear because, apart from the caveat around the records mentioned above, the modelled summer signal would be unlikely to resemble a summer-biased annual record. It seems more likely that it might look like a weighted combination of two seasons. Given this, I think the main finding around seasonality could benefit from further elaboration.*

**Authors' reply:**

We thank the reviewer for raising this and will elaborate on our findings, especially the potential summer bias, in more detail in the revised version of the manuscript. In our study, we compare the modelled summer signal with the reconstructed annual signal as well as the modelled annual signal with the reconstructed annual signal. Under the assumption that the reconstructed annual signal is biased towards the summer months, we would assume that it shows a higher agreement with the modelled summer signal than with the modelled annual signal. As stated above, we aim to discuss the seasonal definitions of the Temp12k dataset in more detailed in a revised version of the manuscript. Together with the above proposed additional tests we are convinced this will substantially improve the manuscript.

**Action:**

- We will add additional analyses as stated above and expand the discussion of seasonality as suggested by the reviewer.

**Reviewer's comment:**

*It's not always clear how the present study's use of the EBM adds to the existing debate. Instead the results seem to highlight where the EBM is significantly different from the other models and these already span a fairly large gradient of complexity. Perhaps one conclusion that could be strengthened is that a model like TransEBM is not greatly informative for this type of problem where seasonal differences are large?*

**Authors' reply:**

We are convinced that simple models like TransEBM are useful tools for testing the sensitivity to individual forcings (e.g. as in Ellerhoff et al., 2022). We will add an analysis of such sensitivity experiments to the revised version, especially testing the role of volcanic forcing as outlined also in respone to reviewer 1. As the reviewer described, TransEBM shows the lowest agreement in temperature trend with the proxy reconstructions from all investigated models. Considering the linear nature of TransEBM, this highlights the fact that the observed temperature trends in the proxy reconstructions cannot be explained by the forcings themselves, but rather that internal, non-linear feedbacks of the climate system are vital. We will discuss the linear response of TransEBM and how its inclusion benefits our study in more detail in the revised version as described in response to reviewer 1 as well.

Action:

- We will include additional sensitivity experiments with TransEBM in the manuscript, analyse and discuss them.

Reviewer's comment:

*The MPI-ESM simulations by Bader et al (2020) show a cooling mode as discussed in your introduction. It would be good to clarify here whether the simulation with MPI-ESM analysed here shows a similar result as it does not look to be the case from figure 1 or B3. If this is not the case, does this support your hypothesis about ocean spin up temperature being important or is there some other reason that can be identified?*

Authors' reply:

Our study provides a comparison of early, mid-, and late Holocene climate trends. However, the simulation by Bader et al. (2020) covers only mid-Holocene to present-day. In our study, we therefore used the MPI-ESM simulation by Kleinen et al. (2020). The simulations differ in resolution, considered forcings and overall simulated time span. While the simulation by Bader et al. (2020) covers the mid-Holocene (6000 BCE) to 1850 CE, the simulation by Kleinen et al. (2020) runs from LGM to present-day and covers the full period of our study. The simulations also differ in applied ice sheet forcings. While the simulation by Kleinen et al. (2020) uses prescribed transient ice sheets from GLAC-1D (Tarasov et al., 2012; Briggs et al., 2014; Ivanovic et al., 2016), the simulations performed by Bader et al. keep ice sheet topography and extent constant.

The late Holocene cooling pattern, which correlates with the sea-ice extent in the simulation by Bader et al. (2020) is not observable in the MPI-ESM simulation by Kleinen et al. (2020) used in this study. Although there are the differences in the model setup and forcing, the lack of cooling mode may be related to the initialization of the simulations at different times and thus might support a stronger imprint of the ocean's initial state on the Holocene trend. We note that the Kleinen et al. simulation was initialized during the LGM and does not show the same cooling trend as the Bader et al. simulation initialized in the early to mid-Holocene, which

shows some cooling. Considering that TransEBM does not show this cooling mode while being in perpetual equilibrium, this observation might be due to the MPI-ESM simulation by Bader et al. being in dis-equilibrium. At the same time, other simulations initialized during that time seem to not show the same cooling trend (c.f. Askær et al. 2022 Fig. 3), although a definite statement on this would require applying the methodology from Bader et al.

More, and more faithful, multi-proxy reconstructions of temperature variables on land, surface and deep ocean would help to further constrain the temperature trends. For example, observations of mean ocean temperature (MOT) showed that MOT peaked at the onset of the early Holocene around 12 thousand years ago and were rather constant afterwards (Bereiter et al., 2018; Haeberli et al., 2021). Yet, for the period of the Holocene climate maximum in global reconstructions around 6 thousand years ago MOT reconstructions are only sparsely available (Bereiter et al., 2018; Baggenstoss et al. 2019; Haeberli et al. 2021) due to clathrate formation in the ice core. To investigate this further, looking into differences of the energy uptake of the ocean during the Deglaciation and the Holocene would be an option if such model output were widely available. Additionally, mid-Holocene ice core data would potentially allow to fill the gap in the reconstructed MOT to clarify if there was a thermal maximum in MOT as well.

**Action:**

- We will enhance discussion regarding the presence/absence of the cooling mode in the two MPI-ESM simulations.

**Reviewer's comment:**

*It seems like a major difference between TransEBM and the other 3D models arises in the polar regions. Could you elaborate on this?*

**Authors' reply:**

TransEBM shows a more homogeneous response during the early Holocene and lacks dynamical features in areas of cooling shown in the other simulations, but overall shows a similar magnitude of warming. TransEBM further shows a later onset of cooling in the high latitudes and the cooling is smaller than in the other models.

The difference between the warming displayed by the different models (Fig. 3 in the original manuscript) during the early Holocene can be largely related to the prescribed land and sea-ice. For TransEBM, the areas of pronounced warming reflect those ice cover changes and linearly related feedbacks, and the prescribed sea-ice extent results in fairly zonal and smooth warming patterns. Regarding the more complex models, dynamic features around the ice sheets and sea-ice/ocean interactions produce cooling/warming features due to atmospheric dynamics. The smaller cooling during the mid- and late Holocene in TransEBM might possibly

be related to smaller and linear feedbacks in response to orbital insolation changes, in comparison to the feedbacks in the more complex models.

Additionally, TransEBM does not model seasonality of sea-ice or apply any land model or land use forcing which would enable it to simulate and respond to changes in the biosphere during that period. Finally, since TransEBM only simulates diffusive heat transport it does not resolve ocean currents. The observed differences – and broad similarities -- between TransEBM and the other three models are a clear indication that it is of importance to take complex feedback mechanisms into account to understand the Holocene climate.

**Action:**

- We will further elaborate on the differences between TransEBM and the more complex models in the polar regions in the manuscript.

**Reviewer's comment:**

*Line 67: "Sea-ice extent is linearly interpolated between the Last Glacial Maximum (LGM) and present-day states given in Zhuang et al. (2017)."*

*Line : "The sea-ice was interpolated using the same method. For sea-ice, the distributions given by Zhuang et al. (2017) were used as fix points for present-day and LGM conditions."*

*Related to the point above, the role that sea-ice plays in the EBM is not clear from the sentences above. I suggest you add a paragraph briefly summarising TransEBM itself (in addition to forcings) to the Appendix.*

**Authors' reply:**

In TransEBM sea-ice is handled as a prescribed transient forcing to the model. As described above, this has the effect that there is no sea-ice feedback. Additionally, no seasonal sea-ice cycle is simulated or prescribed. A detailed description of how the sea-ice forcing was generated as well as a general description of TransEBM is given by Ziegler and Rehfeld (2021). However, we agree with the reviewer that our manuscript would benefit from extending our summary of TransEBM in the Appendix, and we are happy to do so in a revised version of the manuscript.

**Action:**

- We will further elaborate on TransEBM in the Appendix as suggested by the reviewer.

**Reviewer's comment:**

*Line 91: Could you spell out in more detail how you extract a seasonal reconstruction from temp12k?*

**Authors' reply:**

The Metadata of the Temp12k dataset contains the information "Seasonality" and "Season General" (Kaufmann et al. 2020b). The "Seasonality" information specifies the time of year a climate variable in the dataset represents and is given as a list of calendar months (Kaufmann et al. 2020b). Accounting for the variety of seasonalities, this is generalized by the "Season General" metadata. As prescribed above "Season General" classifies seasonality as either annual, summer or winter (Kaufmann et al. 2020b). In our study we used this classification made by Kaufmann et al. and directly extracted it from the Temp12k dataset. As discussed above, we will significantly revise and extend the discussion of this in the revised version of the manuscript.

**Action:**

- We will further elaborate on the assumptions and we will further test the effect of the assigned seasonalities on our results as described above.

**Reviewer's comment:**

*Is JJA the best choice of season for the southern hemisphere? Related to this, can you replace summer with northern hemisphere summer (JJA) in the rest of the text.*

**Authors' reply:**

We thank the reviewer for pointing out that our usage of "summer" and "JJA" was unclear in the manuscript. We clarify this in the revised manuscript.

**Action:**

- We will clarify the usage of "summer" and "JJA" in the manuscript.

**Reviewer's comment:**

*Line 129: expect for TransEBM which shows a high latitude cooling and warming over the North Atlantic and the North American East Coast.*
*Can you discuss why TransEBM has hte opposite sign over the ice-sheet?*

**Authors' response:**

While the Laurentide ice sheet is still retreating during the early Holocene in all simulations, the impact of the decreasing northern summer (c.f. Fig 1 k) insolation at high latitudes is largest in TransEBM. This signal is then superimposed on the retreat of the ice sheet leading to a net negative temperature trend for the Hudson Bay. However, we can partly observe a warming at the north American east coast where the warming in response to the glacial retreat is dominant. Overall, this points to a lack of dynamical features in TransEBM.

Action:

- Will discuss this difference between TransEBM and the other models.

**Reviewer's comment:**

*Line 228: Another relevant reference here is Dallmeyer et al (2022): https://doi.org/10.1038/s41467-022-33646-6.*

**Authors' reply:**

We agree with the reviewer that this is a valuable reference for our statement and will add this in the revised version.

**Action:**

- We will include the reference.

**References:**

Askjær et al., 2022. Multi-centennial Holocene climate variability in proxy records and transient model simulations. Quat. Sci. Rev. https://doi.org/10.1016/j.quascirev.2022.107801

Bader, J. et al.: Global temperature modes shed light on the Holocene temperature conundrum, Nature Communications, 11, https://doi.org/10.1038/s41467-020-18478-6,2020.

Baggenstos, D. et al.: (2019). Earth's radiative imbalance from the Last Glacial Maximum to the present. In Proceedings of the National Academy of Sciences (Vol. 116, Issue 30, pp. 14881–14886)

Bereiter, B., Shackleton, S., Baggenstos, D., Kawamura, K., and Severinghaus, J.: Mean global ocean temperatures during the last glacial transition, Nature, 553, 39–44, https://doi.org/10.1038/nature25152, 2018.

Briggs, R. D., Pollard, D., and Tarasov, L.: A data-constrained large ensemble analysis of Antarctic evolution since the Eemian, Quaternary Sci. Rev., 103, 91–115, https://doi.org/10.1016/j.quascirev.2014.09.003, 2014.

Ellerhoff, B., Kirschner, M. J., Ziegler, E., Holloway, M. D., Sime, L., & Rehfeld, K. (2022). Contrasting State-Dependent Effects of Natural Forcing on Global and Local Climate Variability. *Geophysical Research Letters*, *49*(10), e2022GL098335. https://doi.org/10.1029/2022GL098335ha

Haeberli, M. et al.: Snapshots of mean ocean temperature over the last 700 000 years using noble gases in the EPICA Dome C ice core, Climate of the Past, 17, 843–867, https://doi.org/10.5194/cp-17-843-2021, 2021

Ivanovic, R. F., Gregoire, L. J., Kageyama, M., Roche, D. M., Valdes, P. J., Burke, A., Drummond, R., Peltier, W. R., and Tarasov, L.: Transient climate simulations of the deglaciation 21–9 thousand years before present (version 1) – PMIP4 Core experiment design and boundary conditions, Geosci. Model Dev., 9, 2563–2587, https://doi.org/10.5194/gmd-9-2563-2016, 2016.

Kaufman et al.: Holocene global mean surface temperature, a multi-method reconstruction approach, Scientific Data, 7, https://doi.org/10.1038/s41597-020-0530-7, 2020a

Kaufman et al.: A global database of Holocene paleotemperature records, Sci. Data, 7, 115, 2020b

Kleinen, T., Mikolajewicz, U., and Brovkin, V.: Terrestrial methane emissions from the Last Glacial Maximum to the preindustrial period, Climate of the Past, 16, 575–595, https://doi.org/10.5194/cp-16-575-2020, 2020

Tarasov, L., Dyke, A. S., Neal, R. M., and Peltier, W.: A data-calibrated distribution of deglacial chronologies for the North American ice complex from glaciological modeling, Earth Plane. Sc. Lett., 315–316, 30–40, https://doi.org/10.1016/j.epsl.2011.09.010, 2012.

Ziegler, E. and Rehfeld, K.: TransEBM v. 1.0: Description, tuning, and validation of a transient model of the Earth's energy balance in two dimensions, https://doi.org/10.5194/gmd-2020-237, 2020

---

## Author Comment (AC3)

**Reply to comments by reviewer 3: Exploring Holocene temperature trends and a potential summer bias in simulations and reconstructions (egusphere-2023-86)**

Christian Wirths, Elisa Ziegler, Kira Rehfeld

**Summary of Changes**

We thank the reviewer for providing thoughtful comments on our work, which we take into account in the revision. In response, we plan to

- substantially add to the analysis to include additional results analysing the uncertainties in the analysis, supporting the robustness of the results we put forward, and analysing the sensitivity of the simulations to different forcings,
- expand the discussion of the results, in particular to further highlight the novelty of our findings,
- discuss the added analyses and experiments and
- extend the literature review further.

Below, we provide detailed responses to the comments and outline the actions we plan to take in response.

**Reviewer's comment:**

This paper uses existing datasets from climate models and proxy databases to investigate Holocene temperature evolution. The paper attempts to compare and contrast these pre-existing data sources but fails to put forth new conclusions or findings. Thus, a major restructuring with new evidence and support would be required for publication of this work. My main reasons for this distinction are as follows:

1. **The paper fails to put forth novel findings:** While it is true that the authors compare and contrast pre-existing modeling and proxy datasets, the findings presented in this paper are not novel in comparison to previous literature. Furthermore, many of the claims made by the paper are not substantiated by robust evidence or are not discussed in adequate detail. Therefore, the paper reads closer to a review paper rather than a journal article.

**Authors' reply:**

We thank the reviewer for their feedback and agree that more detail will help to highlight the novelty of our study. We further agree that the manuscript can be improved by including more of the robustness tests that we performed alongside with the results we already provided, but did not include in the paper so far. As outlined in response to reviewers 1 and 2, we will also further probe the seasonality categorization in the proxy data, the influence of forcings applied

in the simulations — in particular of volcanism — by including sensitivity experiments and overall expand the discussion of uncertainties in support of our results. These results will be incorporated into the existing structure of the manuscript. We are aware that our study, like many others, is not solving the conundrum and is instead excluding explanations that have been suggested in the literature. However, comparing temperature trends from climate models of different complexity and including an unpublished simulation with proxy reconstructions to resolve (dis-)agreement between them in time (by investigating the temperature trend agreement for different parts of the Holocene) and space (by investigating different latitudinal bands) provides a novel perspective on Holocene temperatures appropriate for a journal article. In summary, we will therefore enhance both the review and the original research aspect of the paper which we find useful and timely to resolve the temperature conundrum in the future.

**Action:**

- We will expand the discussion of our methodology and results, in particular with respect to the novelty of our analyses and findings, the robustness of our results and the uncertainties involved.
- We will include further evidence in support of the robustness of our results by including sensitivity experiments that show the impact of modelled forcings on Holocene temperature trends. Further, we will add to the discussion of seasonality by testing the uncertainty attached to the categorization in the proxy database as outlined in response to reviewer 2.

**Reviewer's comment:**

2. **Use of TransEBM is not justified:** The authors highlight simulations with TransEBM, a simple energy balance model that has previously been published, as their primary contribution in the modeling space. However, it is never discussed why a low complexity model like TransEBM is suitable or advantageous for the type of analysis conducted here. In fact, as a reader I am left convinced that use of this model is entirely unnecessary as it underperforms the three other models, two of which are even out of date at this point in time.

**Authors' reply:**

We agree TransEBM does not reproduce reconstructed temperature trends better than many of the other models used. However, the added value of low complexity models, especially when used in addition and comparison to higher complexity models, does not rely on them outperforming the other models. Rather, their strengths lie in their simplicity. For the purpose of this study, the main advantage is that we completely understand the response of the model, e.g., we know that TransEBM is always essentially at equilibrium and that it captures the linear climate response without memory effects. As such, it allows for disentangling the climate signal and understanding how simulated dynamics change with level of complexity (c.f. Held 2005). This simplicity of the model is the reason for its inclusion, a point which we will highlight and

strengthen more in the revision of the manuscript. To this end, we will add the results of our single forcing sensitivity studies to offer further insight into the interplay between forcings, feedbacks and the climate system as discussed in response to reviewers 1 and 2. These sensitivity tests explore, among others, the role of greenhouse gases, sea and land ice, and volcanism. The latter we will contrast to simulations with HadCM3 simulations that we will also add to the manuscript as outlined in response to reviewer 1. This will help highlight the benefits of including simulations from a model of low complexity, as we agree with the reviewer that this was insufficient in the original version of the manuscript.

Action:

- We will include sensitivity studies with TransEBM to explore impact of forcing and the linear climate response further.
- Throughout the manuscript, we will discuss the reasons for including simulations of varying complexity and in particular TransEBM more clearly.

Reviewer's comment:

3. **More thorough literature review must be done:** In the concluding remarks, the paper makes strong claims, for example that climate models are likely missing boundary conditions such as land cover. Previous studies have investigated these topics at length (for example, see Thompson et al., 2022, Science Advances); however, there is no mention of these articles here. The authors should do a more thorough literature review of the state of science surrounding the Holocene temperature conundrum before attempting to publish on this topic.

Authors' reply:

We agree with the reviewer that a stronger discussion of current literature addressing the impact of different forcings onto the Holocene climate system can improve our manuscript and happily include the suggested and related literature on this topic in a revised version of the manuscript.

Action:

- We will enhance the literature review and the discussion of our results in context of the literature.

Reviewer's comment:

4. **Analysis of seasonal bias is not robust:** The analysis of seasonal bias in this paper is performed in a simple manner – annual or summer modelled results are compared with annual or summer proxies. Given the statements made about seasonal bias in proxies, a more thorough analysis that considers each proxy and the inherent assumptions present must be done in order to make the claims made by the authors.

Authors' reply:

Our study so far focusses on the comparison of temperature trends in simulations and proxy reconstructions at the proxy sites individually and throughout the different parts of the Holocene. However, we agree with the reviewer the manuscript can gain additional purpose by expanded regarding the robustness of our results and that this will require providing the results of additional tests. Therefore, we will add further new tests investigating the robustness of the seasonal classification of the proxy reconstructions as outlined in detail in response to reviewer 2. First, we will include simulated and reconstructed winter temperatures in the analysis and compare them with each other, as well as with annual and summer temperatures. Second, we will provide upper bounds for the involved uncertainties by comparing simulated temperatures from different seasons. Lastly, the database labels as annual proxy timeseries that cover more than six months in any given year and include June (Kaufmann et al. 2020a and b). We will test the effect of this in the model simulations by contrasting such timeseries with the full annual timeseries at the proxy locations. This will strengthen the analysis provided, which uses the assessment of the recorded season of authors of the Temp12k database, and test the inherent assumptions of our analysis.

Action:

- We will include further analyses exploring the robustness of our results and the seasonality in the proxy records.
- We will expand the discussion of uncertainties in our analysis and strengthen the discussion of seasonality.

References:

Held, I. M. (2005). The gap between simulation and understanding in climate modeling. *Bulletin of the American Meteorological Society*, *86*(11), 1609–1614. https://doi.org/10.1175/BAMS-86-11-1609

Kaufman et al.: Holocene global mean surface temperature, a multi-method reconstruction approach, Scientific Data, 7, https://doi.org/10.1038/s41597-020-0530-7, 2020a

Kaufman et al.: A global database of Holocene paleotemperature records, Sci. Data, 7, 115, 2020b